# Effects of a forefoot strengthening protocol on explosive tasks performance and propulsion kinetics in athletes: a single-blind randomized controlled trial

Romain Tourillon [1,2*], François Fourchet[2,3], Pascal Edouard[1,4], Jean-Benoît Morin[1,5]

**1** University Jean Monnet Saint-Etienne, Lyon 1, University Savoie Mont-Blanc, Interuniversity Laboratory of Human Movement Sciences (EA 7424), Saint-Etienne, France, **2** Physiotherapy Department and Motion Analysis Lab, Swiss Olympic Medical Center, La Tour Hospital, Meyrin, Switzerland, **3** SFMKS Lab, French Sport Physiotherapy Association, Paris, France, **4** Department of Clinical and Exercise Physiology, Sports Medicine Unit, University Hospital of Saint-Etienne, Faculty of Medicine, Saint-Etienne, France, **5** Sports Performance Research Institute New Zealand (SPRINZ), Auckland University of Technology, Auckland, New Zealand

* romain.tourillon@univ-st-etienne.fr

## Abstract

### Purpose

To investigate the effects of an 8-week "periodized high-load" forefoot strengthening protocol on athlete's metatarsophalangeal joints (MTPj) flexion torque, MTPj flexors volume, sprint acceleration, cutting, and jumping overall performance and kinetics.

### Methods

Twenty-height highly-trained athletes were randomized into a TRAINING or control group. Following a 4-week control period, TRAINING performed an 8-week forefoot strengthening protocol (2 sessions per week) followed by a 4-week detraining period. CONTROL group athletes were asked to continue their usual activities. During weeks 1, 5, 14 and 18, we assessed MTPj flexion torque, MTPj flexors volume, maximal sprint acceleration, 90-degree cutting, vertical and horizontal jumps, and foot-ankle hops. A linear mixed model was used along with individual statistical analyses using the minimal detectable change (MDC).

### Results

TRAINING significantly and substantially increased MTPj flexion torque and MTPj flexors volume (effect size [ES]: 1.36–1.96; p < 0.001) with 92% of athletes exceeding the MDC. Subsequently, TRAINING induced significant improvements in cutting and horizontal jumping performance (ES: 0.53–1.14; p < 0.01) with 42–67% of athletes exceeding the MDC. These gains were partly attributed to enhanced medio-lateral

**Data availability statement:** All relevant data are within the manuscript and its Supporting Information files.

**Funding:** Financial support was obtained for this project by the University of Saint-Etienne and Saint-Etienne Métropole AAP Recherche 2022.

**Competing interests:** No authors have competing interests.

ground reaction force transmission during cutting and increased propulsive horizontal force production and transmission during jumping (ES: 0.38–0.57; p < 0.05). Despite no effects on overall sprint acceleration performance, vertical propulsion kinetics at maximal speed improved in TRAINING after intervention (ES: 0.87–1.19; p < 0.01). No significant differences were found between the results of the interventional and detraining period demonstrating potential long-lasting effects.

## Conclusion

An 8-week "periodized high-load" forefoot strengthening protocol allowed to improve MTPj maximal torque and MTPj flexors volume. This strength gains led also to cutting, horizontal jump overall performance and kinetics improvement as well as greater maximal speed propulsion kinetics. MTPj strength capacity may exert a more substantial impact on performance and kinetics on horizontally and medio-lateral-oriented explosive movements than on vertically-oriented ones.

---

## 1. Introduction

The human foot is a complex structure that comprises multiple joints and degrees of freedom, and plays a crucial role in modulating lower limb energetics [1,2]. Recent biomechanical studies have shown that the foot can dissipate ~18% and generate ~12% of the net center of mass energy during rapid tasks such as running, jumping or hopping [1–3]. Beyond its contribution to lower limb energetics, the foot functions as an efficient/modulating lever during propulsion, facilitating the rapid transfer of force from ankle plantar flexors to the ground through to its forefoot region [1,2,4,5]. Notably, studies have shown that the reduction in metatarsophalangeal joint (MTPj) stiffness results in decreased positive foot-ankle power during the push-off phase of rapid tasks [1–3]. These findings suggest that both the extrinsic foot muscles toe flexors (EFMtf), including the *flexor hallucis longus* and *flexor digitorum longus*, and the intrinsic foot muscles (IFM) actively stiffen the forefoot by generating moments around the MTPj. This action accelerates the center of mass and enhances lower limb force transmission to the ground during propulsion [4,6]. Moreover, MTPj strength has been shown to be moderately correlated (r = 0.38–0.55) with performance in explosive tasks such various vertical jumps [7], agility tests [8] and more recently, propulsion kinetics at maximal speed during sprinting [9]. For these reasons, the potential importance of a "stronger" forefoot to athletic performance enhancement has been widely argued [10–12].

However, evidence remains contradictory, and questions persist regarding whether a "stronger" forefoot can indeed enhance sport performance and propulsion kinetics in explosive tasks. Although several intervention studies have demonstrated increased toes or MTPj flexion strength following various strength training protocols [11–15], these gains have not consistently translated into overall performance in explosive tasks, and not in highly-trained populations. For instance, vertical jump height has improved after 6 and 12 weeks of forefoot strengthening protocols in

some studies [12,13], while no significant changes were observed after 4, 7 and 8 weeks in others [11,14,15]. Notably, all studies reported improvements in the horizontal jump length [11–14], suggesting a potentially greater impact of forefoot strengthening on horizontally oriented explosive movements compared to vertically oriented ones. Regarding sprint acceleration, one study reported an improvement in 50-m dash sprint time after 8 weeks of toe flexion strength training [14]; however the results used a stopwatch for performance evaluation, which may introduces significant sources of error and limit accuracy. Conversely, a more recent study found no improvement in sprint performance and ground reaction force (GRF) kinetics after 4 weeks of strength training [15]. As for cutting tasks, no study to our knowledge has investigated the effects of a forefoot strengthening protocol on cutting time, despite the moderate correlation found between these features [8].

Given these inconsistencies, it is possible that the generic protocols and exercises used in these studies, such as the *"short foot, tower curl"* or *"sitting toes isometric flexion"*, while effective in eliciting some MTPj' strength gain (~+30%), may not be sufficiently relevant for enhancing MTPj biomechanical function during explosive movements [11,16]. Moreover, most studies included moderately trained participants; with strengthening exercises and strength evaluation performed on the same ergometer; this led to a potential bias toward greatest strength and athletic performance improvement. Indeed these protocols design and exercises were shown as being not as effective for trained and highly-trained athletes [15,16]. Therefore, this study aims to investigate the effects of an 8-week forefoot strengthening protocol using a "periodized higher-load approach" on (1) MTPj flexion torque (primary outcome), (2) MTPj flexors cross-sectional area (CSA), (3) foot morphology, (4) sprint acceleration, cutting and jumping overall performance, and (5) GRF propulsion kinetics in highly-trained athletes. Based on our clinical experience, and some previous results, we hypothesized that our forefoot strengthening protocol would induce significant MTPj strength gains and MTPj flexors muscle hypertrophy. Furthermore, consistent with the literature, we hypothesized, that these strength gains would positively affect horizontal jumping performance [11–14], as well as propulsion kinetics at maximal speed during sprint acceleration [9].

## 2. Materials and methods

### 2.1. Overall study design

This study was a single-center, single-blind randomized controlled clinical trial reviewed and approved by the Committee for the protection of persons (CPP Ouest III – Poitiers, number: 2022-A00376-37) and registered on ClinicalTrials.gov (Identifier: NCT05574322). To produce the present manuscript, we used the Consolidated Standards of Reporting Trials (CONSORT) [17] (Fig 1). The overall study procedure design is presented in Fig 2 and details of the intervention and outcomes measurements are presented in the paragraphs below. The entire study (intervention, outcomes and data collection) was conducted at the University of Saint-Etienne within the Interuniversity Laboratory of Human Movement Sciences from January 2023 to June 2024 and all participants gave written informed consent before the beginning of the study.

### 2.2. Participants

Athletes were recruited from sport science university community and internal network by using infographics and newspaper advertisements. Athletes were eligible for inclusion if they met the following criteria: (1) age between 18 and 40 years and (2) practicing at a regional or national level in the following sports: soccer, rugby, track and field, basketball, handball, volleyball or tennis. Exclusion criteria were: (1) reported foot or ankle pain within the previous 6 months, (2) leg or foot fracture within the past year or severe foot deformity, (3) history of at least one surgery to the lower limb within the past 6 months, (4) prior foot strengthening experience in the past 6 months, (5) contraindication to neuromuscular electrical stimulation (NEMS) such as pacemaker, seizure disorder, or pregnancy and (6) score of Foot Ankle Ability Measure for sport subscale <80% and Cumberland Ankle Instability Tool ≤23 points [18,19].

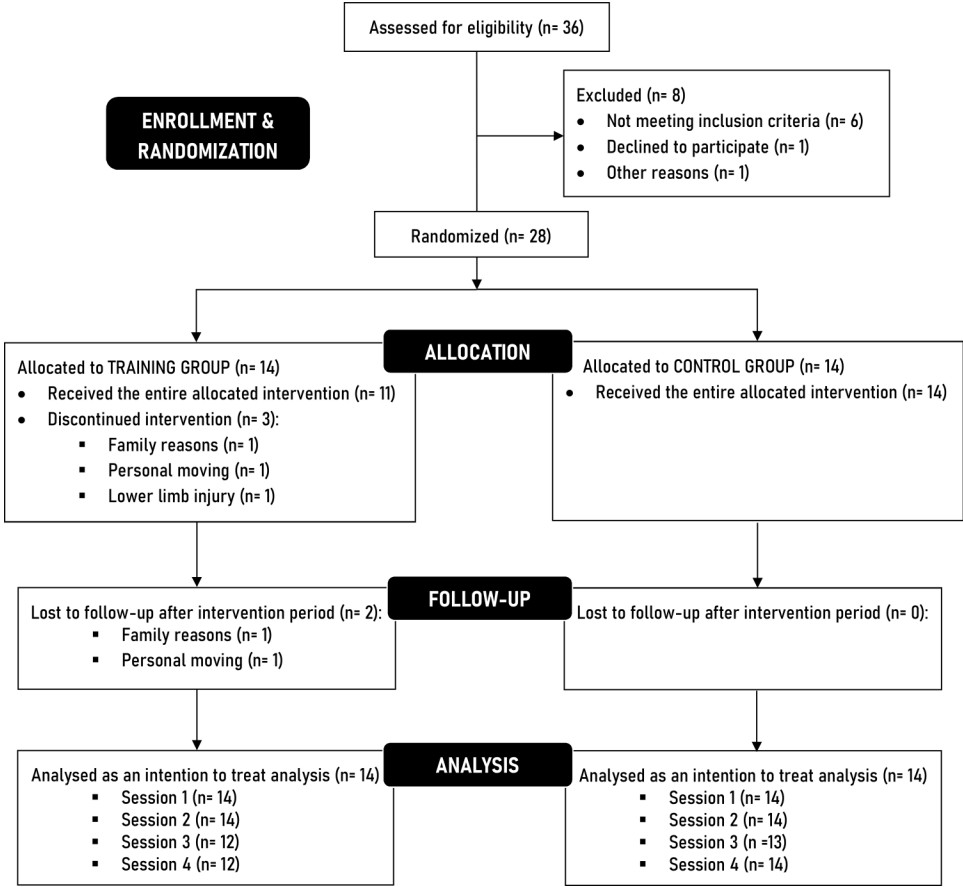

**Fig 1. CONSORT (Consolidated Standards of Reporting Trials) flow diagram illustrating the flow of participants through the study.**

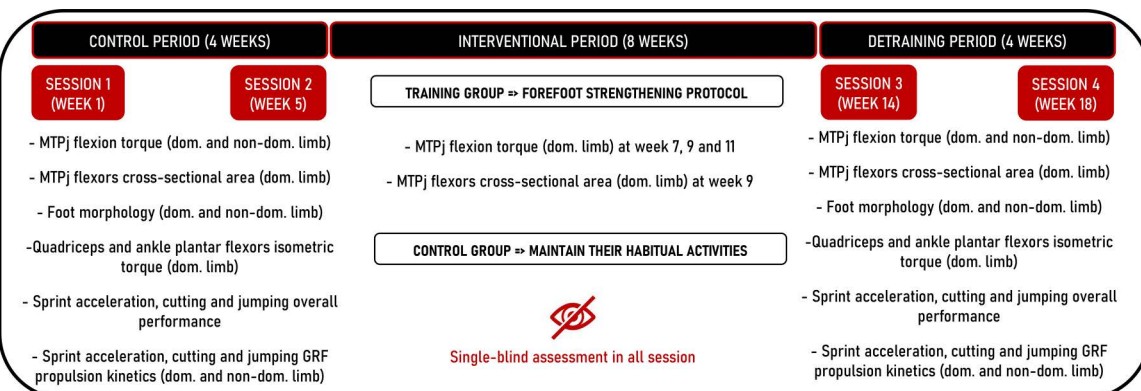

**Fig 2. Forefoot strengthening protocol intervention design.** (MTPj, metatarsophalangeal joints; dom, dominant; non-dom, non-dominant; GRF, ground reaction forces).

## 2.3. Randomisation and blinding

After the session 2 (during week 5) at the end of baseline/control period, athletes were randomly allocated using a random permuted block randomization (1:1 training group allocation) into: 1) forefoot strengthening group (TRAINING) or 2) a control group (CONTROL) (Fig 1). The randomization sequence was prepared at the University Hospital of Saint-Etienne by an independent operator who generated the allocation sequence using REDCap web application. The operator (RT) who performed all the data collection was blinded to the intervention allocation until the end of the study.

## 2.4. Intervention

The participants in the TRAINING group undertook an 8-week forefoot strengthening protocol while continuing their usual sporting training routine. The protocol consisting of performing each week: one supervised session in a gym of ~35-min with at least one strength and conditioning student (sometimes two) and one unsupervised session at home of ~30-min (16 sessions in total). Each week the two sessions were separated by 2 or 3 days. Detailed program and exercise execution for the forefoot strengthening protocol are provided in S2 Fig. Forefoot strengthening protocol supporting information. In brief, athletes followed a multi-components protocol with 5 exercises aimed at enhancing the "propulsing foot". The supervised session included 3 exercises: "forefoot iso-push + NMES", "1st ray dynamic iso-hold" and "forefoot rebound jumps" following resistance training fundamentals (e.g., progressive overload, variation, specificity, volume, intensity) adjusted every two weeks in training blocks. The unsupervised session comprised 2 exercises performed at home: "foot-bridge iso-hold + NMES" and "forefoot iso wall push" " for which participants were provided an NMES device to use throughout the protocol [20]. Compliance and fidelity were monitored by filling a training log with compliance defined as the proportion of prescribed exercises completed, and fidelity as the extent to which participant executed the prescribed exercises, sets, repetitions and target loads [21,22]. Participants rated the difficulty of each session using a perceived exertion scale (RPE) out of 10 (Borg's CR-10) [23]. Participants in the CONTROL group did not perform (and were not informed about) the forefoot strengthening protocol, and were instructed to maintain their regular sporting activities (weekly sport volume) during the intervention period.

## 2.5. Outcomes

**2.5.1. Procedures.** At the beginning of the first session (week 1), limb dominance was determined using three unskilled tasks and three skilled tasks [24] whereas our primary outcome (MTPj maximal isometric flexion torque) was then assessed across seven time points: 1) pre-training at the beginning of the baseline/control period (week 1); 2) pre-training at the end of the 4-week baseline/control period (week 5); 3,4,5) three times during the 8-week training period (week 7), week 9 and week 11); 6) post-training at the end of the training period (week 14) and 7) post-training at the end of the 4-week detraining period (week 18). In parallel, our secondary outcomes (MTPj flexors CSA, foot morphology, sprint acceleration, cutting and jumping overall performance and kinetics) were assessed across four time points at week 1, week 5, week 14 and week 18 (Fig 2). Baseline values of all primary and secondary outcomes were calculated as the mean of session 1 (week 1) and session 2 (week 5) and were compared to session 3 (week 14) to assess the acute effects, and to session 4 (week 18) to evaluate post-detraining effects (Fig 2).

**2.5.2. Primary outcome: Metatarsophalangeal joint (MTPj) maximal isometric flexion torque.** MTPj maximal isometric flexion torque was assessed with the MTPj in approximately ~30° dorsiflexion using a custom-built dynamometer equipped with a 6-component force sensor (Nano 25, ATI Industrial Automation, Garner, NC). This dynamometer showed a good test-retest reliability (ICC = 0.85 [95%CI: 0.73, 0.92]) for measuring maximal isometric MTPj flexion torque in healthy athletes [25]. Following the methodology of a previously protocol [9], athletes were instructed to "push as hard" as possible during five 5-second contractions with a 1-minute rest interval between each effort. Trials were excluded if heel detachment or knee extension was observed (examiner decision). If the coefficient

of variation (CV) among the three highest attempts exceeded 5%, additional contractions were required. Force signal was recorded at 1000 Hz using a power-lab data system (16/30-ML880/P, ADInstruments, Bella Vista, Australia), visualized via Labchart software (v7, ADInstrument) and analyzed using a custom-written Python code. The highest force value from the five contractions was considered for further analyses as the "MTPj maximal isometric pushing force" on the z-axis, "MTPj maximal isometric gripping force" on the y-axis, and "MTPj maximal isometric total force" on the yz-resultant time series. The corresponding torques around the MTPj were assessed by multiplying the absolute force by the length of the first ray (total foot length minus truncated foot length) using the Arch Height Index Measurement System (JAKTOOL Corporation, Cranberry, NJ) [26]. Additionally, maximal isometric quadriceps (MIQt) and ankle plantar flexors torque (MIPFt) were measured using an isokinetic dynamometer (Con-Trex MJ, CMV AG, Dubendorf, Switzerland) with data visualization and analysis conducted following previously mentioned procedures [9]. These evaluations were done only on the dominant side for time constraint and were aimed to monitor the evolution of maximal strength in other muscular groups along the forefoot strengthening protocol as they may also influence force production and transmission in the ground.

**2.5.3. Secondary outcomes: MTPj flexors cross-sectional area (CSA) and foot morphology.** Ultrasound scans were conduct by an experienced operator using a portable musculoskeletal ultrasound system (Esaote, My Lab Sigma, Serie 7410, Genova, Italy) equipped with a 4–15 MHz (maximum depth 5 cm) wideband linear array probe. Due to the presence of multiple IFM and EFMtf (~14 muscles) and time constraint, the cross-sectional area of the *abductor hallucis* (AbH) and the *flexor digitorum longus* (FDL) was assessed as they represent both IFM and EFMtf muscle group respectively. Each athlete laid in supine position with the foot-ankle in a stable neutral position at zero degrees and the posterior aspect of the knee supported in approximately 15 degrees of flexion [27]. Muscle location was determined based on established protocols [27,28]. The CSA of the AbH was obtained along a scanning line perpendicular to the long axis of the foot at the anterior aspect of the medial malleolus. The CSA of FDL was imaged on a transverse line drawn at 50% of the distance between the medial tibial plateau and inferior border of the medial malleolus on the medio-posterior aspect of the tibia. Care was taken to maintain good contact between the probe and skin without applying excessive pressure. All ultrasound images were saved, decoded, and measured by the same operator using Image J software (National Institute for Health, Bethesda, MD, USA). The mean of three measurements were taken on each site (AbH, FDL) and retained for statistical analysis with the probe repositioned between each recording. Foot posture was assessed in a relaxed bipedal standing position using the Foot Posture Index–6 item version (FPI-6) [29], while the foot morphological deformation was assessed using the Arch Height Index Measurement System [26]. Consistent with previous methodologies, 1 and 2-dimensional foot morphological deformation was assessed using "navicular drop", "arch height flexibility" and "foot mobility magnitude" [30].

**2.5.4. Secondary outcomes: Sprint acceleration, cutting and jumping tasks.** Kinetic data for all explosive tasks were collected using a segment of ~5.4-m (6x900 mm) force platforms (Kistler, Winterhur, Switzerland) embedded in the laboratory ground and interfaced with a laptop running BioWare software (version 5.11, Kistler Instruments Inc., Amherst, NY, USA), with a sampling rate of 2000 Hz. Standardized shoes with a minimalist index of 95% (Saguaro™) were provided to minimize the effect of varying shoe material properties on the performance and kinetic variables measured. The detailed protocol for these tasks is available in S3 File. Study protocol details supporting information and has been previously described [9]. For the sprint acceleration task, participants performed maximal accelerations under three conditions [31]: "high-acceleration", "medium-acceleration", and "low-acceleration", allowing for the analysis of GRF over the 0–6 m, 7–13 m and 30–36 m sections, respectively [31]. For the cutting task, participants were instructed to "run as fast as possible" for 5 m, make a single and complete foot contact with the force platform, execute a 90° turn, and then sprint "as fast as possible" again for an additional 5 m to the finish line. The jumping tasks involved performing a "vertical and horizontal CMJ", with participants instructed to "jump as fast and as high as possible" or "as fast and as far as possible", while keeping their hands on their hips throughout. The best performance

trial for each explosive task was retained for statistical analysis. Performance metrics included the best sprint and cutting time, recorded using timing gates system (Microgate, Bolzano, 113 Italy), jump height for "vertical CMJ" based on the impulse-momentum relationship [32] and jump length for "horizontal CMJ". GRF signals were filtered using a Butterworth low-pass digital filter at a cutoff frequency of 50 Hz. A custom MATLAB script (Mathworks, R2022b, Natrick, MA, USA) was used to compute discrete GRF variables as for previous sprint [31,33], cutting [34,35] and jumping [32,36] biomechanics studies. These variables, assessed during the contact phase for sprinting and cutting, and the propulsive phase for jumping, included (a) the impulse (integral over time) of the vertical (Fz) and/or horizontal (Fy) and/or medio-lateral GRF (Fx); (b) the effective impulse of the vertical GRF, i.e., the product of the stance phase duration by the average vertical GRF applied above body-weight; [c] the net impulse of the horizontal GRF and the impulse of the positive (propulsive) component of the horizontal GRF; (d) the ratio of forces (RF), i.e., the ratio of the step- or phase averaged Fy or Fx component divided by the resultant of the step- or phase averaged GRF (FTot); and (e) the contact time defined by the events of foot-strike and toe-off from the raw GRF data (Fz threshold of 20 N for sprinting and 10 N for cutting). Finally, participants performed a jumping task called "Foot-Ankle Rebound Jumps" (FARJ) [37,38]. This test showed excellent test-retest reliability (ICC = 0.92 [95%CI: 0.81, 0.96]) for assessing foot-ankle reactive strength metrics and stretch-shortening cycle (SSC) capacities in athletes [39]. Athletes were instructed to jump "as high as possible" while keeping their lower limbs fully extended, and to push against the ground "as quickly as possible" with only a plantarflexion of the ankle and the MTPj during eight jumps [37,38]. From the contact and flight time measured with an optoelectronic system (Optojump Next, Microgate, Bolzano, 113 Italy) the mean reactive strength ratio of four jumps (excluding the first and last two jumps of the series) was calculated as it showed the highest acceptable reliability and variability in athletes [39].

**2.5.5. Sample size.** The sample size was determined based on power calculations: using a conservative mean effect size (Cohen's d = 1.84) derived from all previously mentioned interventional studies [11–15]. For the primary outcome (MTPj flexion torque) a total sample size of 18 participants (N = 9 per group) was required. Then, with a 5% significance level, a power of 90% and accounting for a 20% drop-out rate, a minimal total of 22 participants (11 per group) was required.

## 2.6. Statistical analyses

Descriptive statistics was applied for all continuous variables, with means and standard deviations reported. Employing an intention-to treat analysis, separate linear mixed effects models (LMMs) were used to evaluate the impact of the TRAINING group on the primary outcome (MTPj flexion torque). Comparable analyses were performed for secondary outcomes related to MTPj flexors CSA, foot morphology, sprint acceleration, cutting and jumping overall performance, and GRF propulsion kinetics in each explosive task. Full factorial models were implemented, incorporating fixed effects for intervention group (TRAINING, CONTROL) and time (baseline, week 14, week 18), along with a random effect to account for between-participant variation. Effect sizes were derived from post-hoc contrasts from LMMs and reported. Additionally, individual-level statistical analysis was employed due to its relevance for sport scientists and practitioners in addition to group-average assessments [40,41]. We further examined whether individual changes in each group exceeded the established minimal detectable change (MDC). MDC at a 95% confidence interval was calculated as: $MDC = TE \times 1.96 \times \sqrt{2}$ where TE is the standard deviation of the differences of outcomes during the baseline period between session 1 (week 1) and session 2 (week 5) divided by the square root of 2: $TE = \frac{SD}{\sqrt{2}}$ [42,43]. Individuals surpassing the MDC positively (increased performance) were classified as "*positive responders*", those exceeding it negatively (decreased performance) were "*negative responders*", while those not surpassing it were "*trivial responders*". The final conclusion of the effects of the forefoot strengthening protocol was drawn at primary and secondary endpoint (week 14 and week 18) from both the linear mixed effects models and individual-level analysis results. The level of significance set at p < 0.05 and all data were analysed using JASP (Amsterdam 0.12.2.0).

## 3. Results

### 3.1. Participants

Of 36 individuals initially interested, a total of 28 athletes met inclusion criteria and were enrolled in this study from January 2023 to June 2024 (Fig 1). Athletes practiced soccer (n = 6), track and field (n = 6), basketball (n = 9), handball (n = 1), volleyball (n = 3) and tennis (n = 3). At baseline, there were no significant differences between TRAINING (n = 14) and CONTROL (n = 14) group for sex, age, dominance, height, weight, BMI, body fat percentage, weekly sport volume or foot-ankle patient-reported outcomes measures (p > 0.05) (Table 1).

Of the 14 individuals randomized to the TRAINING group, 2 athletes were unable to complete the entire study protocol and were lost to follow-up after intervention resulting in a 14% drop-out rate whereas one athlete reported pain/discomfort during a session (Fig 1).

### 3.2. Compliance

TRAINING group achieved a compliance level of 94.3 ± 23.3% and 92.3 ± 27.9% for the supervised and unsupervised sessions respectively, with fidelity rates of 88.4 ± 24.5% and 92.3 ± 28.0%. Mean RPE levels were reported as 5.3/10 for supervised sessions and 4.5/10 for unsupervised sessions.

### 3.3. Primary outcome: MTPj maximal isometric flexion torque

The results of linear mixed model and individual responses results are summarized in Table 2. No significant differences were observed between sessions for the CONTROL group (p > 0.05) (Table 2). In contrast, the athletes in the TRAINING

**Table 1. Study participant characteristics at baseline.**

|  | Training (n = 14) | | Control (n = 14) | | p-value* |
|---|---|---|---|---|---|
|  | N | (%) | N | (%) |  |
|  | Mean | ±SD | Mean | ±SD |  |
| Gender |  |  |  |  |  |
| Female | 3 | (21.4%) | 4 | (28.6%) | 0.663 |
| Male | 11 | (78.6%) | 10 | (71.4%) |  |
| Age (years) | 23.1 | ±4.9 | 21.6 | ±2.7 | 0.377 |
| Height (cm) | 179.9 | ±11.0 | 177.8 | ±9.2 | 0.592 |
| Mass (kg) | 72.8 | ±9.8 | 71.5 | ±9.2 | 0.719 |
| BMI (kg/m²) | 22.5 | ±1.8 | 22.6 | ±1.7 | 0.885 |
| Body fat percentage (%) | 17.4 | ±5.5 | 19.2 | ±4.3 | 0.343 |
| Dominance |  |  |  |  |  |
| Right | 3 | (21.4%) | 2 | (14.3%) | 0.622 |
| Left | 11 | (78.6%) | 12 | (85.7%) |  |
| Weekly sport volume (hours/week) | 9.6 | ±4.9 | 7.2 | ±3.4 | 0.139 |
| FAAM sport dominant foot (%) | 99.7 | ±0.8 | 99.1 | ±1.8 | 0.558 |
| FAAM sport non-dominant foot (%) | 98.5 | ±3.1 | 98.7 | ±3.3 | 0.949 |
| CAIT score (dominant foot) | 29.0 | ±2.4 | 28.5 | ±2.4 | 0.332 |
| CAIT score (non-dominant foot) | 28.6 | ±2.0 | 27.4 | ±3.4 | 0.230 |

BMI, Body Mass Index; FAAM, Foot and Ankle Ability Measure; CAIT, Cumberland Ankle Instability Tool.

*Independent t tests or Wilcoxon rank sum tests were used based on normal distribution (Shapiro-Wilk test). Fisher exact test was used to identify differences in distribution of the dichotomous data.

**Table 2. Results of linear mixed model and individual responses comparing groups [TRAINING versus CONTROL] after post-training and post-detraining period for relative metatarsophalangeal joints maximal isometric flexion torque and MTPj flexors cross-sectional area (CSA).**

| Variable | Group | S1 (week 1) & S2 (week 5) | S3 (week 14) | S3 vs Baseline (Adjusted mean difference) | | Between-Group differences | Individual responses | S4 (week 18) | S4 vs Baseline (Adjusted mean difference) | | Between-Group differences | Individual responses |
|---|---|---|---|---|---|---|---|---|---|---|---|---|
| | | Mean ±SD | Mean ±SD | Δ ± [95% CI] | ES + [95% CI] | ES + [95% CI] | Pos/ Triv/ Neg [% pos] | Mean ±SD | Δ ± [95% CI] | ES + [95% CI] | ES + [95% CI] | Pos/ Triv/ Neg [% pos] |
| **DOMINANT FOOT** | | | | | | | | | | | | |
| Relative pushing torque (Nm/kg) | Training | 0.25±0.08 | 0.33±0.07 | **0.08 [0.06; 0.09]\*\*\*** | **1.80 [1.42; 2.17]** | **1.44 [1.04; 1.81]\*\*\*** | **11/1/0 [92%]** | 0.33±0.07 | **0.07 [0.06; 0.09]\*\*\*** | **1.70 [1.30; 2.08]** | **1.36 [0.98; 1.74]\*\*\*** | **11/1/0 [92%]** |
| | Control | 0.25±0.05 | 0.25±0.06 | −0.00 [−0.02; 0.02] | −0.01 [−0.38; 0.35] | | 1/11/1 [8%] | 0.25±0.05 | −0.00 [−0.02; 0.01] | −0.02 [−0.38; 0.33] | | 1/11/2 [7%] |
| Relative gripping torque (Nm/kg) | Training | 0.08±0.03 | 0.10±0.04 | **0.02 [0.00; 0.03]\*** | **0.43 [0.11; 0.78]** | 0.25 [−0.13; 0.62] | 0/12/0 [0%] | 0.10±0.04 | **0.02 [0.00; 0.03]\*** | **0.49 [0.13; 0.84]** | 0.29 [−0.08; 0.66] | 0/12/0 [0%] |
| | Control | 0.07±0.02 | 0.08±0.03 | 0.01 [−0.00; 0.02] | 0.25 [−0.13; 0.63] | | 0/13/0 [0%] | 0.08±0.03 | 0.01 [−0.01; 0.02] | 0.22 [−0.16; 0.60] | | 0/14/0 [0%] |
| Relative total torque (Nm/kg) | Training | 0.26±0.08 | 0.35±0.07 | **0.07 [0.06; 0.09]\*\*\*** | **1.80 [1.42; 2.15]** | **1.36 [1.00; 1.74]\*\*\*** | **11/1/0 [92%]** | 0.34±0.07 | **0.07 [0.06; 0.09]\*\*\*** | **1.70 [1.35; 2.08]** | **1.34 [0.98; 1.72]\*\*\*** | **11/1/0 [92%]** |
| | Control | 0.26±0.05 | 0.26±0.05 | 0.00 [−0.01; 0.02] | 0.07 [−0.28; 0.43] | | 1/11/1 [8%] | 0.26±0.05 | 0.00 [−0.01; 0.02] | 0.01 [−0.33; 0.35] | | 1/11/2 [7%] |
| Abductor hallucis CSA (cm²) | Training | 2.27±0.27 | 2.49±0.23 | **0.18 [0.15; 0.21]\*\*\*** | **2.27 [1.89; 2.65]** | **1.88 [1.51; 2.24]\*\*\*** | **11/1/0 [92%]** | 2.50±0.23 | **0.19 [0.16; 0.22]\*\*\*** | **2.41 [2.03; 2.78]** | **2.13 [1.75; 2.51]\*\*\*** | **11/1/0 [92%]** |
| | Control | 2.21±0.23 | 2.18±0.24 | −0.02 [−0.05; 0.01] | −0.24 [−0.60; 0.11] | | 0/13/0 [0%] | 2.19±0.24 | −0.02 [−0.05; 0.01] | −0.32 [−0.70; 0.05] | | 0/13/1 [0%] |
| Flexor Digitorum Longus CSA (cm²) | Training | 1.77±0.26 | 1.94±0.26 | **0.18 [0.15; 0.21]\*\*\*** | **2.60 [2.22; 2.99]** | **2.03 [1.66; 2.41]\*\*\*** | **11/1/0 [92%]** | 1.95±0.27 | **0.19 [0.17; 0.22]\*\*\*** | **2.78 [2.40; 3.15]** | **2.12 [1.75; 2.50]\*\*\*** | **11/1/0 [92%]** |
| | Control | 1.66±0.14 | 1.66±0.15 | −0.00 [−0.03; 0.02] | −0.06 [−0.42; 0.31] | | 0/12/1 [0%] | 1.66±0.13 | −0.00 [−0.03; 0.03] | 0.01 [−0.38; 0.38] | | 0/14/0 [0%] |
| **NON-DOMINANT FOOT** | | | | | | | | | | | | |
| Relative pushing torque (Nm/kg) | Training | 0.23±0.07 | 0.31±0.07 | **0.07 [0.06; 0.09]\*\*\*** | **2.02 [1.67; 2.40]** | **1.96 [1.56; 2.34]\*\*\*** | **11/1/0 [92%]** | 0.30±0.07 | **0.07 [0.06; 0.08]\*\*\*** | **1.89 [1.54; 2.27]** | **1.61 [1.23; 1.98]\*\*\*** | **11/1/0 [92%]** |
| | Control | 0.23±0.06 | 0.22±0.06 | −0.01 [−0.02; 0.01] | −0.19 [−0.54; 0.16] | | 0/12/1 [0%] | 0.23±0.06 | 0.01 [−0.01; 0.01] | 0.06 [−0.35; 0.47] | | 1/11/2 [7%] |
| Relative gripping torque (Nm/kg) | Training | 0.07±0.03 | 0.09±0.03 | **0.02 [0.01; 0.03]\*\*** | **0.68 [0.30; 1.02]** | **0.60 [0.22; 0.94]\*\*** | 3/9/0 [25%] | 0.10±0.03 | **0.03 [0.02; 0.03]\*\*\*** | **0.94 [0.57; 1.29]** | **0.88 [0.54; 1.26]\*\*\*** | 3/9/0 [25%] |
| | Control | 0.08±0.02 | 0.08±0.01 | −0.00 [−0.01; 0.01] | −0.04 [−0.38; 0.30] | | 0/13/0 [0%] | 0.07±0.02 | −0.01 [−0.01; 0.01] | −0.11 [−0.49; 0.23] | | 1/12/1 [7%] |

*(Continued)*

Table 2. (Continued)

| Variable | Group | S1 (week 1) & S2 (week 5) | S3 (week 14) | S3 vs Baseline (Adjusted mean difference) | | Between-Group differences | Individual responses | S4 (week 18) | S4 vs Baseline (Adjusted mean difference) | | Between-Group differences | Individual responses |
|---|---|---|---|---|---|---|---|---|---|---|---|---|
| | | *Mean ±SD* | *Mean ±SD* | *Δ ± [95% CI]* | *ES + [95% CI]* | *ES + [95% CI]* | *Pos/ Triv/ Neg [% pos]* | *Mean ±SD* | *Δ ± [95% CI]* | *ES + [95% CI]* | *ES + [95% CI]* | *Pos/ Triv/ Neg [% pos]* |
| Relative total torque (Nm/kg) | Training | 0.24±0.08 | 0.32±0.07 | **0.08 [0.06; 0.09]**\*\*\* | **2.05 [1.67; 2.43]** | **1.96 [1.56; 2.34]**\*\*\* | **11/1/0 [92%]** | 0.32±0.07 | **0.08 [0.06; 0.09]**\*\*\* | **2.02 [1.65; 2.40]** | **1.72 [1.35; 2.13]**\*\*\* | **11/1/0 [92%]** |
| | Control | 0.24±0.06 | 0.23±0.06 | −0.01 [−0.02; 0.01] | −0.16 [−0.54; 0.19] | | 0/12/1 [0%] | 0.24±0.06 | 0.00 [−0.01; 0.01] | 0.03 [−0.30; 0.38] | | 1/11/2 [7%] |

ES, Effect Size; Pos, Positive; Triv, Trivial; Neg, Negative;

*p<0.05;

**p<0.01;

***p<0.001.

group significantly increased MTPj maximal isometric flexion torque of their dominant and non-dominant foot for pushing and total torque with large between-group effect size post-training period (ES ≈ 1.36–1.96; p<0.001) and post-detraining period (ES ≈ 1.34–1.72; p<0.001), with improvements exceeding 1.6 to 2.5 times MDC scores, respectively (Table 2, Fig 3). Individuals' responses analysis indicated that 11 out of 12 (92%) athletes were classified as *"positive responders"* across this primary outcome after both post-training and detraining periods (Fig 3).

### 3.4. Secondary outcome: MTPj flexors CSA

No significant differences were observed between sessions for the CONTROL group (p>0.05) (Table 2). Regarding MTPj flexors CSA, athletes in the TRAINING group significantly increased their AbH CSA and FDL CSA on their dominant foot with large between group effect size after post-training period (ES ≈ 1.36–1.96; p<0.001) and post-detraining period (ES ≈ 1.34–1.72; p<0.001) with improvements exceeding 2.9 to 3.4 times MDC scores, respectively. Individuals' responses analysis indicated that 11 out of 12 (92%) athletes were classified as *"positive responders"* across this secondary outcome after both post-training and detraining periods (Fig 4). No significant differences were detected between sessions or groups for foot morphology, MIQt, and MIPFt variables in either foot (Table 2 in S4 File).

### 3.5. Secondary outcomes: Sprint acceleration, cutting and jumping overall performance

The main results from the linear mixed model and individual response analysis are presented in Table 3, with secondary findings detailed in Table 3 in S4 File supporting information. No significant differences between sessions were observed for the CONTROL group in any secondary outcomes. In sprinting performance, the TRAINING group showed reductions in 10, 17 and 34-meter sprint times; however, these decreases were not statistically significant and fell below the MDC thresholds, with only 0–33% of participants classified as *"positive responders"* post-intervention (Table 3 in S4 File). In cutting performance, the TRAINING group significantly reduced 90° cutting times for both dominant and non-dominant feet, with moderate between-group effect sizes post-training (ES=0.53; p<0.01) and post-detraining (ES=0.61–0.65; p<0.001), though these reductions did not surpass MDC scores (Figs 5 and 6). Post-training, 33% of athletes were positive responders for the dominant foot, increasing to 42% post-detraining; for the non-dominant foot, the rates were 50% and 58%, respectively. Concerning jumping performance, the TRAINING group significantly increased vertical CMJ jump

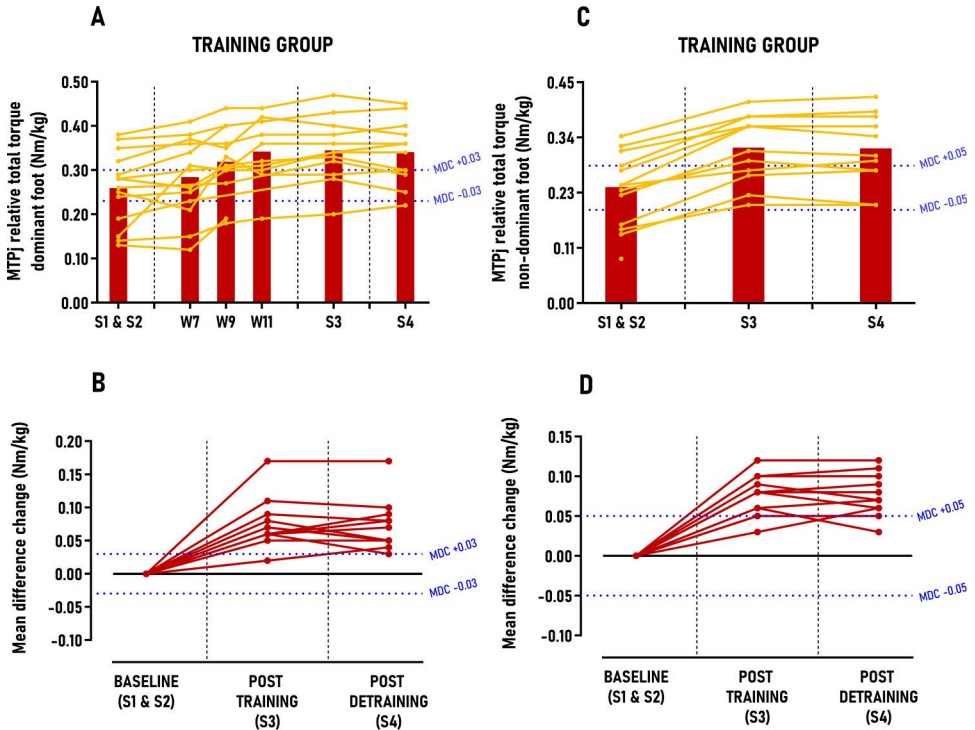

**Fig 3. Group average and individual participant data displayed in time-series graphs and individual mean difference changes in comparison to the minimal detectable changes (MDC) post-training and detraining period.** Subgraphs A&B) MTPj relative total torque – dominant foot (Nm/kg) and C&D) MTPj relative total torque – non-dominant foot (Nm/kg).

height and FARJ reactive strength ratio, with small to moderate between-group effect sizes (ES = 0.44–0.71; p < 0.05), yet these gains did not exceed MDC scores (Fig 1 in S4 File) with only 0–17% of *"positive responders"* identified post-intervention (Table 3 in S4 File). Conversely, horizontal CMJ jump length showed significant increases with large between-group effect sizes (ES = 1.03–1.14; p < 0.001) post-training and detraining, surpassing MDC scores by 1.5 to 1.1 times, respectively (Fig 7). *"Positive responders"* accounted for 67% post-training and 58% post-detraining (Table 3).

### 3.6. Secondary outcomes: Specific GRF propulsion kinetics

Significant results from the linear mixed model and individual responses are presented in Table 3. In sprinting kinetics, the TRAINING group significantly increased relative effective vertical impulse at maximal speed (30–36 m), with large between-group effect sizes (ES = 0.87–1.19; p < 0.001) post-training and detraining, surpassing MDC scores by 1.0 to 1.3 times (Table 3). Three out of four athletes (75%) were classified as *"positive responders"* after the intervention (Table 3). In cutting kinetics, the TRAINING group significantly increased their medio-lateral force ratio for both dominant and non-dominant feet, with small effect sizes for the dominant foot (ES = 0.39–0.41; p < 0.05) and moderate effect sizes for the non-dominant foot (ES = 0.59–0.75; p < 0.01), although these improvements did not exceed MDC scores (Figs 5 and 6). Between 33% and 58% of athletes were classified as *"positive responders"* after the intervention, with approximately 56% of these athletes also improving their cutting time performance. In jumping propulsion kinetics, the TRAINING group significantly increased their relative concentric horizontal impulse and horizontal force ratio, with moderate to large effect sizes (ES = 0.57–1.13; p < 0.01), yet these gains did not surpass MDC scores (Fig 7; Fig 2 in S4 File). Between 25 and 42% of athletes were classified as *"positive responders"*, with approximately 72% of these athletes also improving their horizontal CMJ jump length performance.

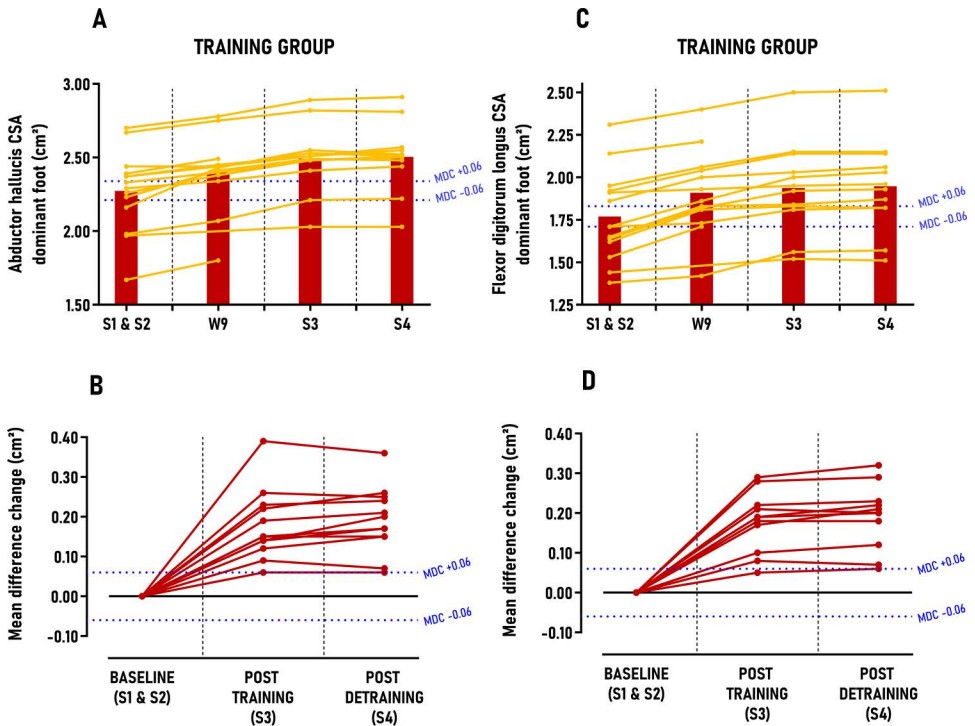

**Fig 4. Group average and individual participant data displayed in time-series graphs and individual mean difference changes in comparison to the minimal detectable changes (MDC) post-training and detraining period.** Subgraphs A&B) Abductor hallucis cross sectional area (CSA) – dominant foot (cm$^2$) and C&D) Flexor digitorum longus cross sectional area (CSA) – dominant foot (cm$^2$).

## 4. Discussion

As hypothesized, our findings revealed that the "periodized higher-load approach" used in this study resulted in substantial and rapid increases in MTPj flexion torque and MTPj flexors CSA, while no changes were observed in foot morphology. Additionally, these strength gains contributed to significant enhancements in cutting and horizontal CMJ performance for a high proportion of *"positive responders"* (42–67%), partly due to increased medio-lateral force transmission to the ground during cutting and improved propulsive horizontal force production and transmission during jumping. Although overall sprint performance and acceleration kinetics remained unchanged post-protocol, we observed an increase in vertical propulsion kinetics at maximal speed, confirming our first hypothesis [9]. Finally, our results indicated that all these improvements were observed at the immediate end of the protocol, but also tended to last after protocol cessation (4 weeks), since no significant differences were found in TRAINING athletes between the training and detraining periods.

### 4.1. MTPj maximal isometric flexion torque and MTPj flexors CSA

The results of this study showed that an 8-week "periodized higher-load approach" effectively increases MTPj maximal isometric flexion torque by approximately 30% in both the dominant and non-dominant foot in highly trained athletes. This finding confirmed that MTPj flexors (EFMtf and IFM) are responsive to high-load exercise protocols, achieving significant strength gains within a short period, even in this trained population. The observed increase of MTPj maximal isometric flexion torque aligns with existing literature, which reports an increase of MTP flexion strength by 14% for Nagahara et al. after 4 weeks [15], 18% for Unger et al. after 6 weeks [13], 32% for Kokkonen et al. after 12 weeks [12], 35% for Hashimoto et al. after 8 weeks [14] and 43% for Goldmann et al. after 7 weeks [11]. While our study's strength gains vary

**Table 3. Main results of linear mixed model and individual responses comparing groups (TRAINING versus CONTROL) after post-training and post-detraining period for overall performance during cutting and jumping and ground reaction force performance kinetics during sprinting, cutting and jumping.**

| Variable | Group | S1 (week 1) & S2 (week 5) | S3 (week 14) | S3 vs Baseline (Adjusted mean difference) | | Between-Group differences | Individual responses | S4 (week 18) | S4 vs Baseline (Adjusted mean difference) | | Between-Group differences | Individual responses |
|---|---|---|---|---|---|---|---|---|---|---|---|---|
| | | Mean ±SD | Mean ±SD | Δ ± [95% CI] | ES + [95% CI] | ES + [95% CI] | Pos/ Triv/ Neg [% pos] | Mean ±SD | Δ ± [95% CI] | ES + [95% CI] | ES + [95% CI] | Pos/ Triv/ Neg [% pos] |
| **OVERALL PERFORMANCE** | | | | | | | | | | | | |
| Cutting time dominant foot (s) | Training | 0.99±0.06 | 0.96±0.07 | **−0.04 [−0,07; −0.00]*** | **−0.41 [−0.78; −0.05]** | −0.20 [−0.58; 0.17] | **4/8/0 [33%]** | 0.94±0.06 | **−0.06 [−0,09;−0.03]*** | **−0.72 [−1.09; −0.34]** | **−0.61 [−0.96; −0.24]** | **5/7/0 [42%]** |
| | Control | 0.99±0.05 | 0.97±0.08 | −0.02 [−0.05; 0.01] | −0.21 [−0.59; 0.16] | | 2/11/0 [15%] | 0.98±0.06 | −0.01 [−0.04; 0.02] | −0.08[−0.45; 0.29] | | 0/13/1 [0%] |
| Cutting time non-dom. foot (s) | Training | 1.04±0.08 | 0.96±0.06 | **−0.08 [−0,11; −0.04]*** | **−0.83 [−1.21; −0.46]** | **−0.53 [−0.90; −0.16]** | **6/6/0 [50%]** | 0.96±0.06 | **−0.08 [−0,11; −0.05]*** | **−0.88 [−1.27; −0.50]** | **−0.65 [−1.01; −0.28]*** | **7/5/0 [58%]** |
| | Control | 1.00±0.07 | 0.99±0.06 | −0.02 [−0.05; 0.02] | −0.19 [−0.56; 0.18] | | 0/13/0 [0%] | 0.99±0.06 | −0.01 [−0.04; 0.02] | −0.13 [−0.52; 0.25] | | 1/13/0 [7%] |
| Horizontal CMJ jump length (m) | Training | 2.08±0.31 | 2.24±0.26 | **0.13 [0.09; 0.17]*** | **1.27 [0.90; 1.65]** | **1.14 [0.77; 1.51]*** | **8/4/0 [67%]** | 2.21±0.27 | **0.09 [0.05; 0.13]*** | **0.92 [0.54; 1.29]** | **1.03 [0.66; 1.39]*** | **7/5/0 [58%]** |
| | Control | 2.12±0.22 | 2.13±0.24 | −0.01 [−0.05; 0.03] | −0.09 [−0.47; 0.28] | | 1/12/0 [8%] | 2.09±0.24 | −0.03 [−0.06; 0.01] | −0.30 [−0.67; 0.07] | | 0/14/0 [0%] |
| **GRF SPRINTING KINETICS** | | | | | | | | | | | | |
| 30–35m effective vertical impulse (BW.s) | Training | 0.11±0.01 | 0.12±0.02 | **0.01 [0.00; 0.02]*** | **0.87 [0.16; 1.58]** | **0.87 [0.24; 1.42]** | **3/1/0 [75%]** | 0.12±0.01 | **0.01 [0.00; 0.02]*** | **1.19 [0.47; 1.90]** | **1.19 [0.55; 1.74]*** | **3/1/0 [75%]** |
| | Control | 0.11±0.02 | 0.11±0.02 | 0.00 [−0.01; 0.01] | 0.01 [−0.63; 0.63] | | 0/4/1 [0%] | 0.11±0.02 | 0.00 [−0.01; 0.01] | 0.04 [−0.55; 0.63] | | 1/5/1 [20%] |
| **GRF CUTTING KINETICS** | | | | | | | | | | | | |
| Medio-lateral ratio of forces -dominant (%) | Training | 38.4±3.2 | 39.9±3.3 | **1.7 [0.1; 3.4]*** | 0.39 [0.02; 0.76] | 0.31 [−0.06; 0.68] | **4/8/0 [33%]** | 40.0±3.5 | **1.8 [0.2; 3.4]*** | 0.41 [0.03; 0.78] | 0.34 [−0.03; 0.71] | **4/8/0 [33%]** |
| | Control | 39.8±2.3 | 40.0±2.5 | 0.2 [−1.4; 1.8] | 0.04 [−0.33; 0.41] | | 1/12/0 [8%] | 40.0±2.4 | 0.2 [−1.4; 1.7] | 0.05 [−0.32; 0.42] | | 1/13/0 [7%] |
| Medio-lateral ratio of forces – non-dom. (%) | Training | 36.7±3.9 | 39.5±4.0 | **2.9 [1.1; 4.7]*** | **0.59 [0.22; 0.96]** | **0.42 [0.05; 0.79]*** | **5/6/1 [42%]** | 40.4±4.8 | **3.8 [1.9; 5.7]*** | **0.75 [0.38; 1.12]** | **0.57 [0.20; 0.94]*** | **7/5/0 [58%]** |
| | Control | 38.6±2.9 | 39.3±2.7 | 0.6 [−1.2; 2.4] | 0.12 [−0.25; 0.49] | | 0/13/0 [0%] | 39.2±2.8 | 0.6 [−1.2; 2.3] | 0.12 [−0.25; 0.49] | | 2/12/0 [14%] |

*(Continued)*

**Table 3.** (Continued)

| Variable | Group | S1 (week 1) & S2 (week 5) | S3 (week 14) | S3 vs Baseline (Adjusted mean difference) | | Between-Group differences | Individual responses | S4 (week 18) | S4 vs Baseline (Adjusted mean difference) | | Between-Group differences | Individual responses |
|---|---|---|---|---|---|---|---|---|---|---|---|---|
| | | Mean ±SD | Mean ±SD | Δ ± [95% CI] | ES + [95% CI] | ES + [95% CI] | Pos/ Triv/ Neg [% pos] | Mean ±SD | Δ ± [95% CI] | ES + [95% CI] | ES + [95% CI] | Pos/ Triv/ Neg [% pos] |
| **GRF JUMPING KINETICS** | | | | | | | | | | | | |
| Horizontal concentric impulse (BW.s) | Training | 0.08±0.01 | 0.09±0.01 | **0.01 [0.00; 0.01]**\*\*\* | **1.13 [0.57; 1.70]** | **0.38 [0.09; 0.76]**\* | 5/7/0 [42%] | 0.09±0.01 | **0.01 [0.00; 0.01]**\*\* | **0.94 [0.38; 1.32]** | **0.47 [0.19; 0.76]**\*\* | 5/7/0 [42%] |
| | Control | 0.08±0.01 | 0.08±0.01 | 0.00 [−0.19; 0.76] | 0.38 [0.02; 0.66] | | 1/12/0 [8%] | 0.08±0.01 | 0.00 [−0.00; 0.00] | −0.05 [−0.57; 0.38] | | 0/13/1 [0%] |
| Concentric ratio of forces (%) | Training | 34.4±3.8 | 36.1±4.1 | **1.9 [0.7; 3.1]**\*\* | **0.57 [0.20; 0.94]** | **0.42 [0.05; 0.79]**\* | 3/9/0 [25%] | 36.6±4.5 | **2.4 [1.1; 3.6]**\*\*\* | **0.72 [0.35; 1.09]** | **0.50 [0.12; 0.87]**\*\* | 4/8/0 [33%] |
| | Control | 36.5±3.7 | 36.9±3.6 | 0.4 [−0.8; 1.6] | 0.13 [−0.24; 0.50] | | 0/12/1 [0%] | 37.1±3.5 | 0.6 [−0.5; 1.8] | 0.21 [−0.16; 0.58] | | 0/14/0 [0%] |

Non-dom., Non-dominant; CMJ, Countermovement Jump; ES, Effect Size; Pos, Positive; Triv, Trivial; Neg, Negative;

\*p<0.05;

\*\*p<0.01;

\*\*\*p<0.001.

from these previous studies, it is worth noting that the latter two [11,14] employed a "testing is training" approach with untrained males, potentially biasing the results toward greater increases. Moreover, our study yielded significant effect sizes (1.36–1.96), exceeded MDC scores (1.6 to 3.4 times), and a high *"positive responders"* rate (92%), all based on the variability measurements of our custom-made ergometer during the control period. Although this allows a descriptive comparison with prior studies, our study's unique protocol design and statistical analysis prevent direct comparisons. Notably, our protocol involved significantly less volume (sixteen 35-min sessions) compared to previous studies, which involved much more sessions/volume (24–36 sessions) and much higher weekly session frequencies (3–4) [11–14]. These observations suggest that the overloading parameters (body over-loading, forward lean, NMES, maximal isometric pushing, etc) and strength training principles (volume, intensity, progression) used in our study were essential to maximize MTPj strength gains in a time-efficient manner. This is further evidenced by our findings that MTPj maximal isometric flexion torque improvement (+17%) was already significant and exceeded the MDC scores by week 9, i.e., the middle of the protocol (Fig 3). Although our protocol focused on "maximal strength" rather than hypertrophy, the TRAINING group also exhibited on the dominant foot a 9.2% increase in AbH and FDL CSA post-training and detraining periods, consistent with previous studies reporting approximately 3.8% to 11.0% increases in AbH and FDL CSA after either 8 weeks of daily walking in minimalist shoes [44] or an 8-week low-load toe flexors strengthening protocol with multiple exercises [44,45]. While the hypertrophy gains we observed were comparable to those reported in earlier studies, it is noteworthy that these studies involved more than twice our session volume (32–56 sessions) and were conducted in untrained individuals, potentially explaining the hypertrophic effects observed in comparison to our population. Indeed, a previous study has showed that IFM thickness was less developed in minimally or untrained male adults in comparison to highly-trained male sprinters [46]. Similar to MTPj maximal isometric flexion torque, CSA improvements (+5%) were significant very

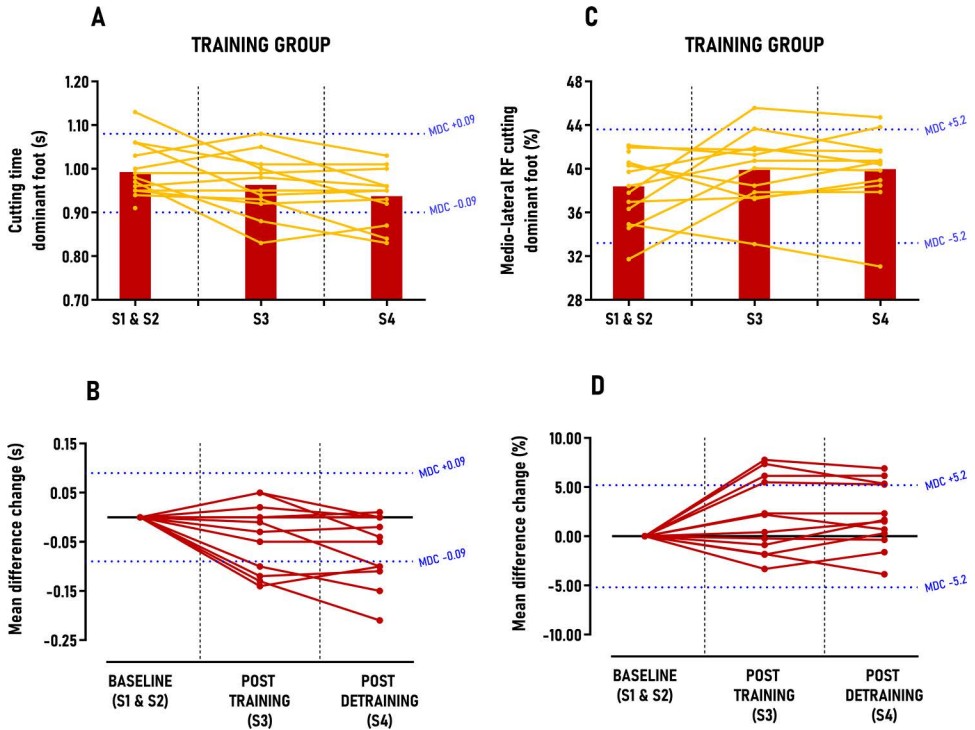

**Fig 5. Group average and individual participant data displayed in time-series graphs and individual mean difference changes in comparison to the minimal detectable changes (MDC) post-training and detraining period.** Subgraphs A&B) Cutting time – dominant foot (s); C&D) Medio-lateral ratio of forces (RF) during cutting – dominant foot (%).

early, (week 9, middle of the protocol) and exceeded the MDC scores (Fig 3). It is well established that strength gains result from a combination of neural and structural factors, with early gains primarily attributed to neural adaptations rather than structural changes [47–49]. Therefore, the parallel increases in MTPj flexion torque and MTPj flexors CSA during the first half of the protocol (+17% vs. +5%) suggest that EFMtf and IFM responded similarly to other lower limb muscles and can be trained accordingly. Furthermore, we demonstrated that these changes in muscle morphology did not lead to alterations in foot morphology (e.g., posture, morphological deformation) (Table 3 in S4 File) which contradicts meta-analyses findings [50–52]. This discrepancy regarding foot posture may be due to the inherent variability and complexity of foot shape and posture among individuals, and, above all, the utopia some authors have found in modifying these factors through foot strengthening [53,54]. Additionally, while the Arch Height Index measurement system is highly reliable for assessing foot length, width, and dorsal arch height [26], the associated equations for calculating "navicular drop", "arch height flexibility", and "foot mobility magnitude" exhibited high variability (coefficient of variation between 18% and 36%) over the study period, complicating accurate interpretation. Finally, although outside the primary scope of this study, the observed improvements in MTPj maximal isometric flexion torque and MTPj flexors CSA may have implications for designing rehabilitation and prevention protocols for foot-ankle complex musculoskeletal disorders, as these parameters are known risk factors for conditions such as plantar heel pain [55] or chronic ankle instability [56,57].

## 4.2. Cutting

In parallel with the enhancements in MTPj flexors strength and CSA, this study is the first to demonstrate that these gains led to significant improvements in cutting performance and propulsion kinetics. Specifically, 40% to 58% of athletes in the

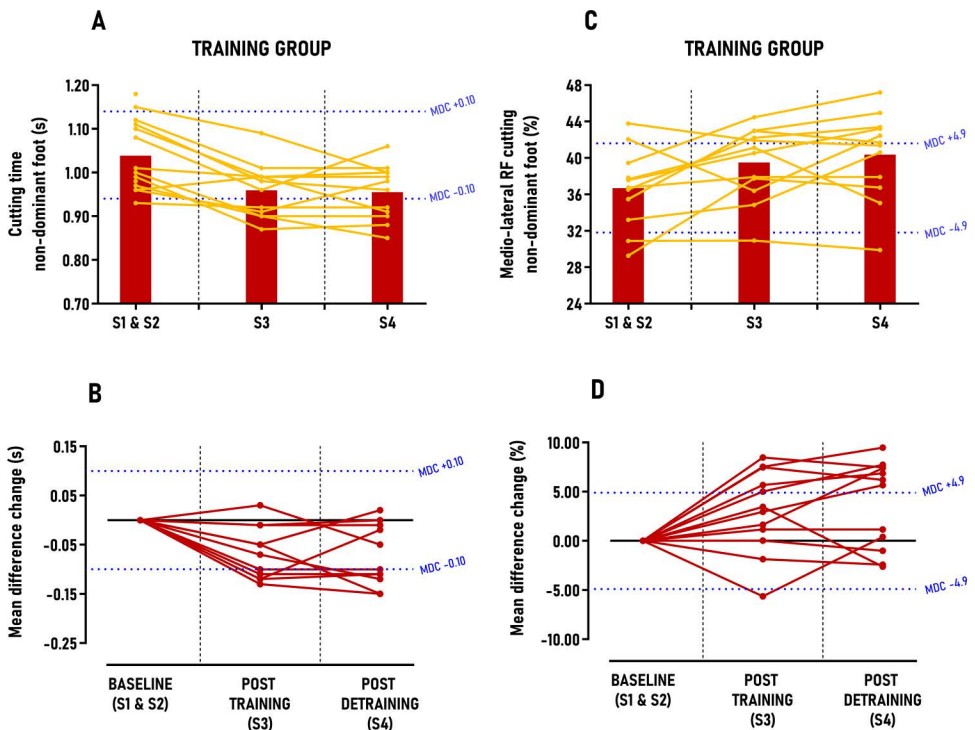

**Fig 6. Group average and individual participant data displayed in time-series graphs and individual mean difference changes in comparison to the minimal detectable changes (MDC) post-training and detraining period.** Subgraphs A&B) Cutting time – non-dominant foot (s); C&D) Medio-lateral RF during cutting – non-dominant foot (%).

TRAINING group were classified as *"positive responders"* post intervention, showing improved performance time with both the dominant and non-dominant foot. This non-hypothesized effect is unprecedented in the literature, which has presented conflicting evidence on this association. For instance, previous research showed a moderate correlation between relative MTPj pushing strength and agility tests performance [8] while a more recent study found no significant correlation between relative MTPj pushing torque and 90° cutting time, but rather with ankle plantarflexors maximal isometric torque and foot-ankle reactive strength [9]. This novel finding might be explained by the fact that the forefoot is the primary zone of peak plantar pressure during side cutting [58], with the MTPj serving as the sole ground contact point for most athletes during the maneuver. Although the foot contributes approximately 14% of the negative work and 3% of positive work during a 45° cutting task [59], our study suggests that improved cutting times in some athletes may result from enhanced medio-lateral force transmission via the MTPj. Indeed, our data revealed that approximately 56% of the *"positive responders"* in cutting performance in the TRAINING group also exhibited medio-lateral force ratios, particularly on the non-dominant foot. Given that peak concentric ankle power and ankle plantar flexors isometric torque have been associated with 75° and 90° cutting time performance [9,60], is it plausible that MTPj strength gains facilitate a better medio-lateral force transmission through the ankle plantar flexors. We hypothesize that this force transmission improvement stems from forefoot strength enhancements rather than ankle plantarflexors, as no changes in MIPFt were observed throughout the study (Table 3 in S4 File). The greater rate of *"positive responders"* in the non-dominant foot could be attributed to limb dominance effects during cutting, as previous research has shown that the non-dominant foot-ankle muscles exhibit less neuromuscular control in the frontal plane during a 90° cutting task compared to the dominant limb [61]. The initially lower MTPj maximal isometric torque and medio-lateral force ratio in the non-dominant foot before the protocol (Tables 2 and

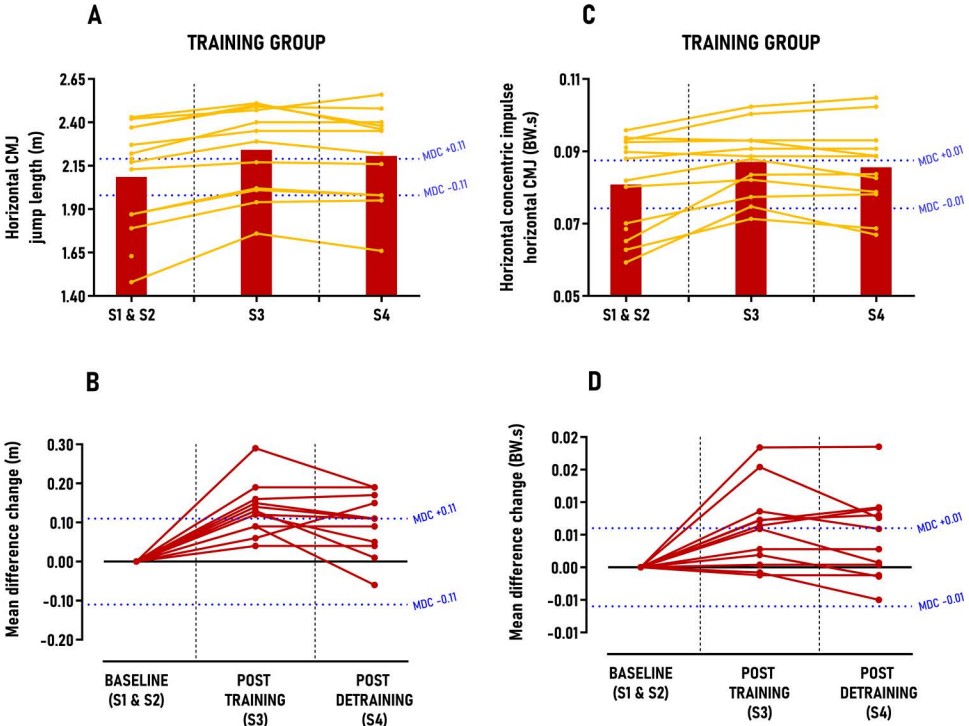

**Fig 7. Group average and individual participant data displayed in time-series graphs and individual mean difference changes in comparison to the minimal detectable changes (MDC) post-training and detraining period.** Subgraphs A&B) Horizontal CMJ jump length (m); C&D) Horizontal concentric impulse – horizontal countermovement jump (CMJ) (BW.s).

3) may have biased these results towards greater improvements. For *"positive responders"* in cutting performance who did not show an increased medio-lateral force ratio, performance improvements may be attributed to enhanced horizontal force production to the ground (e.g., relative horizontal impulse), while one athlete improved cutting performance without any significant GRF changes.

### 4.3. Horizontal jumping

In addition to cutting performance, the second major finding of this study was the improvement in horizontal CMJ performance, with 58% to 67% of athletes in the TRAINING group classified as *"positive responders"*. This finding supports our initial hypothesis and aligns with existing literature, where all authors reported improvements in horizontal jump distance after 7–12 weeks of toe flexors strengthening protocol [11–14]. The impact of MTPj strength on horizontal jump length, rather than vertical CMJ jump height, can be explained biomechanically as athletes must quickly move the trunk ahead of the feet during horizontal CMJ, resulting in MTPj dorsiflexion angle and moments that are more than two and three times greater than during vertical CMJ push-off [11]. For optimal performance, the athlete's body must lean forward during the horizontal jump, shifting the center of force application anteriorly under the toes and increasing the external MTPj moment arm [11,62]. This forward lean causes MTPj flexors to counteract the dorsiflexion moment, maintaining the body's segments above the feet in an upright position and enabling them to operate within an optimal force-length relationship [63,64]. Notably, the importance of body forward lean, combined with body overloading, in increasing MTPj flexors torque and activity was emphasized in our protocol exercises instructions (S2 Fig). It is therefore plausible that post-intervention MTPj strength gains allowed athletes to lean forward more, optimizing their take-off angle for increased jump distance.

Beyond potential biomechanical enhancement of MTPj function and body orientation, our results suggest that horizontal CMJ jump length improvements may also stem from enhanced concentric horizontal force production and transmission for some athletes. Specifically, our findings indicate that approximately 76% of *"positive responders"* in horizontal CMJ performance also showed increases in either horizontal concentric impulse or concentric force ratio. As previously mentioned, improved forward body lean during the push-off phase of the horizontal CMJ may allow better body orientation for generating more concentric horizontal force through the MTPj or transmitting more force horizontally, resulting in higher horizontal take-off velocity. We believe that this force production and transmission do not result from quadriceps or ankle plantar flexors strength gains, as MIQt or MIPFt remained unchanged throughout the study (Table 3 in S4 File). For *"positive responders"* in horizontal CMJ performance who did not increased GRF during the propulsive phase, performance improvements may be attributed to enhanced lower limb joints coordination during the flight phase, a key performance indicator for jump length [65]. Collectively, this study provides novel evidence of the importance of MTPj in horizontally-oriented jumps rather than vertically-oriented ones.

### 4.4. Maximal speed

Finally, the last major finding of this study was the improvement of relative effective vertical impulse at maximal sprinting speed (30–35 m) for a high proportion of *"positive responders"* (75%) in the TRAINING group, despite no overall sprint time performance enhancement. This finding supports our primary hypothesis, as recent biomechanical research demonstrated that MTPj maximal isometric pushing torque and foot-ankle reactive strength are moderately associated with relative effective vertical impulse and contact time at maximal speed rather than with acceleration GRF features [9]. These results have significant implications, as previous studies have highlighted the importance of generating large vertical forces over short contact times during the maximal speed phase to achieve higher speeds [33]. This study and the present findings suggest that greater MTPj strength is crucial for effectively generating high vertical force despite the absence of improvement in foot-ankle reactive strength (Table 3 in S4 File). This supports the importance of training these two physical capacities separately, since previous research showed they were not correlated [9]. Our results could be explained by the *flexor hallucis longus* (largest volume of the EFMtf and IFM) maximal strength improvement inducing a better force transfer from the proximal shank to the distal part of the foot while working in a near-isometric manner [6]. As there is an opposite plantarflexion at the ankle and dorsiflexion the MTPj during the push-off phase [6], the importance of *flexor hallucis longus* maximal strengthening was also emphasized in our protocol with a *"1st ray dynamic iso-hold"* exercise (S2 Fig). However, while this GRF feature improved potentially the maximal speed on the 30–35 m section, it did not translate into an overall 34-m sprint time improvement (Table 3). This discrepancy could be due to the athletes' sprinting 30 meters before reaching the force platforms, which may "dilute" kinetic performance and associated sprint speed during a large portion of the task. These findings contrast with previous studies reporting a 50-m dash sprint time improvement after an 8-week toe flexors strength protocol [14] and no sprinting GRF kinetics improvement during a 50-m sprint [15]. The discrepancies might be explained by the use of a stopwatch sprint time evaluation in the first study, which introduces significant sources of error, and the averaging of all the sprinting GRF variables over the entire 50-m sprint in the latter study, preventing phase-specific analysis as conducted in our study. Future interventional studies are needed to further elucidate the influence of a well-designed MTPj flexion strengthening protocol on sprinting kinetics and kinematics.

### 4.5. Training implications

The results discussed above suggest two significant training implications. Firstly, the observed improvements appear to have not only an acute impact but also a long-lasting effect of up to 4 weeks, as evidenced by the lack of significant differences between the training and detraining periods. Notably, certain performance and GRF variables (e.g., effective vertical impulse at maximal speed) even showed further increases post-detraining compared to the post-intervention period (Table 3). This suggests a sustained learning effect in utilizing the MTPj and forefoot region, even weeks after the protocol

ended, as has been previously demonstrated following a single session of foot electrostimulation under the medial arch [66]. Additionally, significant and rapid improvement in MTPj flexion torque and MTPj flexors CSA can be achieved within just 4 weeks, not only after 8 weeks, when employing a "periodized higher-load approach" in forefoot strengthening protocol. These findings are particularly relevant in the "real world" and in the current context, where time efficiency is paramount when working in athletic development for competitive trained athletes.

### 4.6. Strengths and limitations

This study is the first in this area to utilize a three-phase design: control, intervention and detraining. The control phase allowed for the measurement of TE and associated MDC across the group, enabling the detection of performance changes at the replicated single-subject level post-intervention. This individual statistical analysis, combined with typical group-style statistical analyses (e.g., linear mixed model), is particularly valuable for sports scientists and practitioners as it helps increase results' accuracy and clarify conclusions [40,41]. Without this more individualized approach, we might have erroneously concluded that our protocol improved vertical CMJ jump height performance, whereas the group's improvement was below the MDC threshold, with no individual *"positive responders"* (Table 3). Thus, similar trends may have been overlooked in previous interventional studies that relied solely on group-average assessments [11–15].

The first limitation of this study was that experimental data for the various sprint acceleration conditions were collected across multiple trials with three distinct starting points, a method previously employed [31] due to the force plates dimensions (see Materials and methods). Nonetheless, the descriptive statistics of mean GRF data during sprint acceleration closely align with data from single trials on a ~52-m force platform segment [33]. Secondly, although ultrasonography is not considered as the gold standard to evaluate MTPj flexors morphology, its reliability and validity have been established elsewhere [27,28]. Moreover, this parameter was assessed only in the dominant foot, which does not allow for comparisons between feet. Thirdly, factors influencing muscle training adaptations, such as nutritional intake, were not controlled for practical reasons and may have impacted some outcomes (e.g., CSA). Fourthly, we assessed maximal isometric contractions at only one specific angle for the quadriceps and ankle plantarflexors due to experimental time constraints. Thus, we cannot rule out that these muscles may have increased their dynamic strength, rather than isometric strength, in some athletes in the TRAINING group, potentially affecting overall performance and kinetics outcomes. Finally, the protocol was applied to a sample of young (≤35 years) trained athletes, predominantly male (75%). Therefore, it is plausible that the effects of this protocol may differ in less skilled, older athletes, or in a more gender-balanced cohort.

## 5. Conclusion

In conclusion, this single-blind randomized controlled trial showed that a "periodized higher-load" foot strengthening approach led to significant and rapid improvements of MTPj flexion torque and MTPj flexors CSA within 4 weeks and larger improvements after 8 weeks. These strength gains enabled a significant number of athletes (*"positive responders"*) to achieve better medio-lateral force transmission, enhancing cutting performance, improved propulsive horizontal force production and transmission, thereby increasing CMJ horizontal performance, and greater vertical propulsion kinetics at maximal speed. This supports the notion that MTPj strength may have a more substantial impact on horizontally- and medio-laterally-oriented explosive movements performance kinetics than on vertically-oriented ones.

## Supporting information

**S1 Checklist. CONSORT checklist.**
(DOC)

**S2 Fig. Forefoot strengthening protocol.**
(PDF)

**S3 File. Study protocol details.**
(PDF)

**S4 File. Supplemental tables and figures.**
(PDF)

## Acknowledgments

The results of this study are presented clearly, honestly, and without fabrication, falsification, or inappropriate data manipulation. We would like to thank all the participants for their participation and Clément Moukoko, Sabri Bouzouik and Thomas Monot for their help on data curation and forefoot strengthening protocol supervision.

## Author contributions

**Conceptualization:** Romain Tourillon, François Fourchet, Jean-Benoît Morin.

**Data curation:** Romain Tourillon.

**Formal analysis:** Romain Tourillon.

**Funding acquisition:** Romain Tourillon, François Fourchet, Jean-Benoît Morin.

**Investigation:** Romain Tourillon, Jean-Benoît Morin.

**Methodology:** Romain Tourillon, François Fourchet, Jean-Benoît Morin.

**Project administration:** Romain Tourillon.

**Resources:** Romain Tourillon, François Fourchet, Pascal Edouard, Jean-Benoît Morin.

**Software:** Romain Tourillon.

**Supervision:** Romain Tourillon, François Fourchet, Pascal Edouard, Jean-Benoît Morin.

**Validation:** Romain Tourillon.

**Visualization:** Romain Tourillon.

**Writing – original draft:** Romain Tourillon.

**Writing – review & editing:** Romain Tourillon, François Fourchet, Pascal Edouard, Jean-Benoît Morin.

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
