## [Decision Letter · Decision Letter 0]

10 Mar 2025

PONE-D-24-49343Effects of a forefoot strengthening protocol on explosive tasks performance and propulsion kinetics in athletes: a single-blind randomised controlled trialPLOS ONE

Dear Dr. Tourillon,

Thank you for submitting your manuscript to PLOS ONE. After careful consideration, we feel that it has merit but does not fully meet PLOS ONE’s publication criteria as it currently stands. Therefore, we invite you to submit a revised version of the manuscript that addresses the points raised during the review process.

We look forward to receiving your revised manuscript.

Kind regards,

Riccardo Di Giminiani

Academic Editor

PLOS ONE

Journal Requirements:

“Financial support was obtained for this project by the University of Saint-Etienne and Saint-Etienne Métropole AAP Recherche 2022.”

“Financial support was obtained for this project by the University of Saint-Etienne AAP 2022.”

“Financial support was obtained for this project by the University of Saint-Etienne and Saint-Etienne Métropole AAP Recherche 2022.”

Reviewers' comments:

Reviewer's Responses to Questions

**Comments to the Author**

1. Is the manuscript technically sound, and do the data support the conclusions?

Reviewer #1: Yes

Reviewer #2: Yes

2. Has the statistical analysis been performed appropriately and rigorously? 

Reviewer #1: Yes

Reviewer #2: I Don't Know

3. Have the authors made all data underlying the findings in their manuscript fully available?

Reviewer #1: Yes

Reviewer #2: Yes

4. Is the manuscript presented in an intelligible fashion and written in standard English?

Reviewer #1: Yes

Reviewer #2: Yes

5. Review Comments to the Author

Reviewer #1: General comments

I firstly wanted to congratulate the authors on the work. The topic of the article is very interesting, potentially applicable in many areas (athletic training, functional recovery, rehabilitation context, and return to sports practice). The authors studied the effects of eight weeks of a strengthening forefoot protocol with a “periodized higher-load approach” , on metatarsophalangeal joints flexion torque (primary outcome) and metatarsophalangeal joints flexors CSA, sprint acceleration, and jumping performance and kinetics parameters. (secondary outcomes); the analysis suggests an increase in primary and secondary outcomes.

The manuscript is well written, the specific hypothesis is well formulated and consistent with current literature. The drawing is well structured for this type of study and all sections are very detailed, as the material and method secctions as well as the results (with related figures and tables). Emphasis to supplementary materials, well organized, clear and very original.

In addition, the statistical analysis is very informative and accurate for the interpreting of the results, and because it considers individual variations it could be very appropriate in field contexts.

However, since I am not an expert in the field, I would like to ask you for some clarification on the methods, and I have some small observations that I think can be addressed to further enhance the understanding of the article.

Minor comments:

Comment 1: In section 2.5.1 (results) it is reported that the participants of the TRAINING group have two weekly sessions (supervised and not held on separate days), I suggest explaining this in the text referring to the hours/ days that occur (also approximately) between the two weekly sessions.

Comment 2: In section 2.4 (intervention). Did the participants in the TRAINING group, during the intervention period, perform only the two training sessions included in the study or did they continue their training routine? Please specify in the text.

Comment 3: Did the training sessions performed without supervision at home have the same duration as the sessions performed with supervision? Please specify the approximate durations of the two weekly training sessions in the text.

Comment 4: In order to provide more information to the readers concerning the participants’ characteristics I think perhaps it is appropriate to determine the normality of the sample distribution by the Shapiro-Wilk test.

Comment 5: On line 314, in the description of Figure 3, please correct the unit of measurement (Nm.Kg or Nm/Kg) consistently with the figure. Please recheck.

Comment 6: For completeness of informatio in the discussion section, I suggest including a brief argumentation regarding FARJ, explaining briefly why this parameter is not statistically varied. How does the MTPj affect ankle stiffness modulation?

Comment 7: Some measurements, which are subject to significant changes, were taken were performed only on the dominant limb (i.e.: MTPj flexors CSA), in light of the fact that at baseline the participants showed no statistical differences between the two limbs in term of FAAM or CAIT score, do you believe that the data collected on the dominant can be transferable to the non-dominant? Perhaps a brief comment could be made in the strengths and limitations section.

Reviewer #2: General Comment

The present work represents an extremely interesting and many times underestimated topic. The foot muscles represent a factor that should always be taken into account when explosive tasks such as those analyzed in this study are required. The proposed training methodology and its strength principles represent a significant strength.

The manuscript is well-written, well-structured, with a consistent hypotesis and useful conclusion. The amount of additional material provided is rich in information and helps the reader learn more about the methodology used.

I have only a few clarification to ask:

Minor Comments:

Comment 1

In line 83, clarifying or expanding the meaning of “training is testing protocol” might help in reading the manuscript.

Comment 2

In the supplemental content (S3 Study Protocol Design), the height of the timing gates is 1.20m. For what reason was this height selected? In a three-point crouched start position, could the variation of the body during a sprint lead to measurement errors?

6. PLOS authors have the option to publish the peer review history of their article (what does this mean? ). If published, this will include your full peer review and any attached files.

**Do you want your identity to be public for this peer review?** For information about this choice, including consent withdrawal, please see our Privacy Policy .

Reviewer #1: **Yes: ** Stefano La Greca

Reviewer #2: **Yes: ** Francesco Farina

---

## [Author Response · Author response to Decision Letter 1]

17 Mar 2025

We thank the reviewers for his/her appreciated comment on our manuscript and we did our best to enhance the understanding of the article by addressing each comment on specific files : "Authors response to reviewer 1 and Authors response to reviewer 2".

---

## [Decision Letter · Decision Letter 1]

12 May 2025

Effects of a forefoot strengthening protocol on explosive tasks performance and propulsion kinetics in athletes: a single-blind randomised controlled trial

PONE-D-24-49343R1

Dear Dr. Tourillon,

We’re pleased to inform you that your manuscript has been judged scientifically suitable for publication and will be formally accepted for publication once it meets all outstanding technical requirements.

Kind regards,

Qichang Mei

Academic Editor

PLOS ONE

Additional Editor Comments (optional):

Reviewers' comments:

Reviewer's Responses to Questions

**Comments to the Author**

1. If the authors have adequately addressed your comments raised in a previous round of review and you feel that this manuscript is now acceptable for publication, you may indicate that here to bypass the “Comments to the Author” section, enter your conflict of interest statement in the “Confidential to Editor” section, and submit your "Accept" recommendation.

Reviewer #1: All comments have been addressed

Reviewer #2: All comments have been addressed

2. Is the manuscript technically sound, and do the data support the conclusions?

Reviewer #1: Yes

Reviewer #2: Yes

3. Has the statistical analysis been performed appropriately and rigorously? 

Reviewer #1: Yes

Reviewer #2: Yes

4. Have the authors made all data underlying the findings in their manuscript fully available?

Reviewer #1: Yes

Reviewer #2: Yes

5. Is the manuscript presented in an intelligible fashion and written in standard English?

Reviewer #1: Yes

Reviewer #2: Yes

6. Review Comments to the Author

Reviewer #1: Dear authors,

I am fulfilled with your responses. I find that the article has improved and also the clarity of the methods. I think the article is ready to be published.

Reviewer #2: I would like to congratulate with the authors on the manuscript.

All comments were adequatly addressed.

7. PLOS authors have the option to publish the peer review history of their article (what does this mean? ). If published, this will include your full peer review and any attached files.

**Do you want your identity to be public for this peer review?** For information about this choice, including consent withdrawal, please see our Privacy Policy .

Reviewer #1: **Yes: ** Stefano La Greca

Reviewer #2: **Yes: ** Francesco Farina

---

## [Editor Report · Acceptance letter]

PONE-D-24-49343R1

PLOS ONE

Dear Dr. Tourillon,

I'm pleased to inform you that your manuscript has been deemed suitable for publication in PLOS ONE. Congratulations! Your manuscript is now being handed over to our production team.

Kind regards,

on behalf of

Professor Qichang Mei

Academic Editor

PLOS ONE